# WHY DID THIS MODEL FORECAST THIS FUTURE? INFORMATION-THEORETIC TEMPORAL SALIENCY FOR COUNTERFACTUAL EXPLANATIONS OF PROBABILISTIC FORECASTS

## ABSTRACT

Probabilistic forecasting of multivariate time series is significant to several research domains where multiple futures exist for a single observed sequence. Identifying the observations on which a well-performing model bases its forecasts can enable domain experts to form data-driven hypotheses about the causal relationships between features. Consequently, we begin by revisiting the question: what constitutes a causal explanation? One hurdle in the landscape of explainable artificial intelligence is that what constitutes an explanation is not well-grounded. We build upon Miller's framework of explanations derived from research in multiple social science disciplines, and establish a conceptual link between counterfactual reasoning and saliency-based explanation techniques. However, the complication is a lack of a consistent and principled notion of saliency. Also, commonly derived saliency maps may be inconsistent with the data generation process and the underlying model. We therefore leverage a unifying definition of information-theoretic saliency grounded in preattentive human visual cognition and extend it to forecasting settings. In contrast to existing methods that require either explicit training of the saliency mechanism or access to the internal parameters of the underlying model, we obtain a closed-form solution for the resulting saliency map for commonly used density functions in probabilistic forecasting. To empirically evaluate our explainability framework in a principled manner, we construct a synthetic dataset of conversation dynamics and demonstrate that our method recovers the true salient timesteps for a forecast given a well-performing underlying model.

## 1 INTRODUCTION

The existence of multiple valid futures for a given observed sequence is a crucial attribute of several forecasting tasks, especially surrounding the dynamics of low-level human behavior. These tasks include the forecasting of trajectories of pedestrians (Huang et al., 2019; Mohamed et al., 2020; Rudenko et al., 2020; Salzmann et al., 2021; Zhang et al., 2019), vehicles (Carrasco et al., 2021; Gilles et al., 2022; Zeng et al., 2020; Zhao et al., 2020), and autonomous robots (Ivanovic et al., 2021; Vemula et al., 2017), or other more general nonverbal cues of humans (Adeli et al., 2020; Barquero et al., 2022; Nguyen & Celiktutan, 2022; Raman et al., 2021; Yao et al., 2018) and artificial virtual agents (Ahuja et al., 2019) in group conversation settings. Consequently, rather than making single (i.e. point) predictions, several machine learning methods in these settings have attempted to forecast a distribution over plausible futures (Mohamed et al., 2020; Raman et al., 2021). In this work, we introduce and address a novel research question towards gaining domain-relevant insights into such forecasts: given a reliable underlying model, how can we identify the observed timesteps that are salient for the model's *probabilistic* forecasts over a particular future window?

### 1.1 NOTIONS OF INTERPRETABILITY IN FORECASTING TASKS & DRAWBACKS

Recently, several works have proposed techniques for making interpretable *non-probabilistic* predictions for point forecasting tasks (Lim et al., 2020; Oreshkin et al., 2020; Pan et al., 2021). The

**Table 1:** Classes of Explanatory Question and the Reasoning Required to Answer. Reproduced from Miller (2019, Table 3)

| Question | Reasoning | Description |
|---|---|---|
| What? | Associative | Reason about which unobserved events could have occurred given the observed events |
| How? | Interventionist | Simulate a change in the situation to see if the event still happens |
| Why? | Counterfactual | Simulating alternative causes to see whether the event still happens |

general approach has been to train architectures to produce forecasts that are not only accurate but also interpretable. However, in what Lipton (2017) terms *The Mythos of Interpretability*, the notion of what renders these models interpretable is often not well-grounded and subject to presenting speculation in the guise of explanation Lipton & Steinhardt (2018). Examples of these operationalizations of interpretability[1] for forecasting tasks include: (i) injecting a suitable inductive bias into the model through a set of basis functions and identifying how they combine to produce an output (Oreshkin et al., 2020) (an approach that has recently been applied to the probabilistic setting as well (Rügamer et al., 2022)); (ii) employing a self-attention mechanism to learn temporal patterns while attending to a common set of features (Lim et al., 2020); and (iii) applying the notion of saliency maps from computer vision (Dabkowski & Gal, 2017) to time-series data as a measure of how much each feature contributes to the final forecast (Pan et al., 2021). A broader review of notions of interpretability across domains is in Appendix A.

Irrespective of the notion of interpretability, these methodologies are underpinned by two common attributes: (a) the interpretability mechanism needs explicit training as part of the model architecture, and (b) what constitutes a *good* explanation is subject to the biases, intuition, or the visual assessment of the human observer (Adebayo et al., 2018; Miller, 2019); a phenomenon we refer to as the *interpretation being in the eye of the beholder*. This is especially true for when saliency maps have been used as tools for post-hoc explanations: the computed map may not measure the intended saliency, and even be independent of both the model and data generating process (Adebayo et al., 2018; Atrey et al., 2020; Lapuschkin et al., 2019). Research on saliency-based methods is further confounded by the lack of a common notion of saliency. As Barredo Arrieta et al. (2020, Sec. 5.3) point out, "there is absolutely no consistency behind what is known as saliency maps, salient masks, heatmaps, neuron activations, attribution, and other approaches alike."

### 1.2 SALIENCY-BASED EXPLANATIONS FOR COUNTERFACTUAL REASONING & DRAWBACKS

Due to the inconsistencies mentioned above, we argue that the premise of what constitutes an explanation needs to be well-grounded within existing frameworks of how people define, generate, and present explanations. Miller (2019) recently turned to the vast body of research on the topic in philosophy, psychology, and cognitive science and highlighted the importance of causality in explanation. Specifically, in Table 1 we reproduce the categorization of explanatory questions he proposed based on Pearl and Mackenzie's *Ladder of Causation* (Pearl & Mackenzie, 2018). Using an abstract notion of an 'event' that needs explaining, Miller argues that the *what*-questions involve associative reasoning. *How* questions require interventionist reasoning to determine the set of causes that, if removed, would prevent the event from happening. The *why*-questions are the most challenging, as they require counterfactual reasoning to undo events and simulate other events. These also require associative and interventionist reasoning.

To apply Miller's (2019) framework to forecasting, consider a model $\mathbf{M}$ that predicts features over a future window $t_{\text{fut}}$ by observing features over a window $t_{\text{obs}}$. We argue that most of the existing interpretability approaches—including the two discussed categories involving the injection of inductive biases and attention-based mechanisms—are associative in nature. For a fixed $t_{\text{fut}}$ and single $t_{\text{obs}}$, they reason about the (unobserved) importance of features over $t_{\text{obs}}$ using model parameters or attention coefficients based on one prediction from $\mathbf{M}$ (the 'event'). In contrast, we argue that the perturbation-based saliency methods have the potential to support counterfactual reasoning. The saliency masks are commonly learned by perturbing different parts of the input over

---

[1]Barredo Arrieta et al. (2020) argue for the importance of distinguishing between interpretability and explainability as different concepts, the latter denoting any active action or procedure taken by a model with the intent of clarifying or detailing its internal functions. From this perspective, what the cited works term interpretability is closer to the notion of explainability.

$t_{\mathrm{obs}}$—thereby simulating alternative 'causes' from $\mathbf{M}$'s perspective—and observing the effect on an error metric (the 'event'). However, while saliency-based approaches can, in theory, be promising in answering such why-questions, we identify several issues with how such methods are being applied for deriving explanations: (i) the feature-level manipulations used to produce saliency maps produce distortion interventions that may be inconsistent with the data generation process (Adebayo et al., 2018; Lapuschkin et al., 2019) and not lie on the real-world data manifold or preserve real-world semantics (Atrey et al., 2020); (ii) the notion of saliency in terms of minimizing an error metric is an arbitrary choice, requiring the ground-truth future that may not be available at test; and (iii) for forecasting applications, the saliency map is explicitly retrained for every single observed-future sequence (Pan et al., 2021) and does not capture salient structural patterns across samples.

### 1.3 OUR APPROACH: BUILDING UPON A UNIFYING FRAMEWORK OF BOTTOM-UP SALIENCY

Our broad conceptual contribution is to link counterfactual reasoning with a principled notion of saliency. In Figure 1 we formally express counterfactual reasoning for explaining forecasts using causal graphs (Pearl, 2009). The generic graph expresses relationships between the random variables prior to training the forecasting model denoted by $M$. The exogenous variable $\epsilon_M$ captures the randomness in the training process and modeling choices including the distribution family for representing probabilistic forecasts. The exogenous variable $\epsilon_H$ captures the the randomness in the human observer's choice of observed and future win-

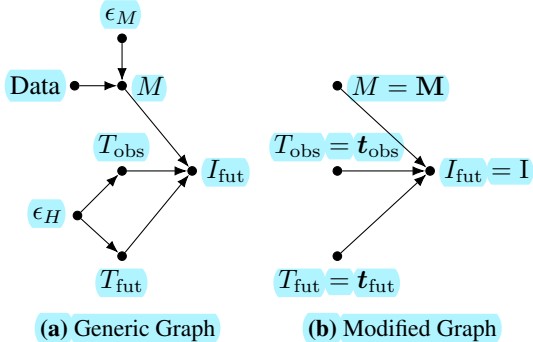

(a) Generic Graph     (b) Modified Graph

**Figure 1:** Causal Graphs for Explaining Forecasts

dows to examine the model. Our central idea is to evaluate the information in the model's predicted distribution denoted by $I_{\mathrm{fut}}$. Specifically, we propose posing the the following counterfactual question: *What information would* $\mathbf{M}$ *have about the future over a given* $t_{\mathrm{fut}}$ *if it observed the features over* $t_{\mathrm{obs}}$? The modified graph for evaluating this question is depicted in Figure 1b. Once the model $\mathbf{M}$ has been trained and the windows $t_{\mathrm{obs}}$ and $t_{\mathrm{fut}}$ have been chosen, the effect of the exogenous variables on the variable $I_{\mathrm{fut}}$ disappears. This yields a deterministic system where the change in the information about the future can be evaluated for different values of $t_{\mathrm{obs}}$ and $t_{\mathrm{fut}}$. Given that the focus of this work is the explanation phase, we assume the modified graph is already available. Note that while the procedure starting from training the forecasting model in the generic graph implicitly follows Pearl's *abduct-action-prediction* process (Pearl, 2009, p.207), estimating the distribution over the exogenous variables from the *abduction* step is conceptually not applicable in this setting.

To relate such counterfactual reasoning to saliency, we propose the following implication:

observing the features at a timestep $t \in t_{\mathrm{obs}}$ results in a change in the model's information about the future $I_{\mathrm{fut}}$ over the given window $t_{\mathrm{fut}} \implies t$ is salient.      (1)

This notion of saliency related to surprise or information associated with observations is grounded in preattentive human cognition (Loog, 2011; vanderHeijden, 1996). Concretely, the statement entails that an observation that changes the certainty of the future is surprising, and thereby salient. Note that the antecedent (on the left of the implication) is a counterfactual statement. The implication entails that for the antecedent to be true $t$ must be salient. However, knowing the antecedent is false is not sufficient to conclude that $t$ is not salient, i.e. there can be other notions of saliency that make $t$ salient. In this work, rather than defining saliency in a top-down manner as a function of some task-specific error metric that cannot be computed at test, as other works have done (Pan et al., 2021), we build our computational framework upon a general expression of bottom-up, or task-agnostic, preattentive saliency (Loog, 2011). Specifically, we derive a closed-form solution for identifying salient timesteps using the differential entropy in $\mathbf{M}$'s forecast over $t_{\mathrm{fut}}$. We then empirically validate the framework using the counterfactual reasoning illustrated in Figure 1 for a forecasting task where human-verifiable ground-truth saliency exists.

Preattentive saliency captures what is perceived to be subconsciously informative before conscious, or attentive, processing by the brain. Here, a surprising or unexpected observation is salient. When applied to forecasting tasks, this idea of surprise (or unexpectedness or informativeness) needs

to be *tied to the future outcome*, which guides our proposal of evaluating the information in the predicted future density. Our approach is counterfactual because we examine changes in this future density in response to *different* realizations of real-world sequences $t_{\text{obs}}$. In contrast, the only existing saliency-based method for time-series forecasting follows associative reasoning within Miller's (2019) categorization: Pan et al. (2021) retrain a saliency map for every *single* $t_{\text{obs}}$ from scratch, one at a time, using a single (point) prediction over the $t_{\text{fut}}$ instead of a distribution.

Our framework addresses all the aforementioned concerns we identified with saliency-based approaches in forecasting: (i) the counterfactuals in our framework pertain to windows of real observed features rather than random perturbations of the input, and therefore preserve real-world semantics; (ii) our information-theoretic perspective of saliency is principled, and given the underlying model, can be used to compute saliency for unseen test data where the ground-truth future is unavailable; (iii) we utilize a distribution over the futures that the model believes could have occurred for a single input: this distribution captures structural predictive relationships between features across multiple samples. Moreover, our framework involves no training for computing the saliency, and can be applied to any underlying model that outputs a distribution over futures.

## 2 BACKGROUND: INFORMATION THEORETIC PREATTENTIVE SALIENCY

Loog (2011) developed a general closed-form expression for saliency using a surprisal-based operational definition of bottom-up attention from the field of computational visual perception. Let $L : \mathbb{R}^n \to \mathbb{R}^d$ be a general $n$-dimensional $d$-valued image. A continuously differentiable feature mapping $\phi : \mathbb{R}^n \to \mathbb{R}^N$ relates every image location $x \in \mathbb{R}^n$ from $L$ to $N$ features. Further, let $p_X$ denote the probability density function over image locations $x$ and $p_\Phi$ denote the probability density function over all feature vectors $\phi(x)$. The distribution $p_X$ captures any prior knowledge that would make one location more salient than another. In the absence of such prior knowledge, $p_X$ is typically chosen to be uniform.

The saliency $S(x)$ of a location $x$ is then defined in terms of the amount of information—or surprise—of its associated feature vector $\phi(x)$ relative to the other feature vectors extracted from the same image. The intuition is that the larger the information content of a certain combination of features, $-\log p_\Phi(\phi(x))$, the more salient the location $x$:

$$S(x) > S(x') \iff -\log p_\Phi(\phi(x)) > -\log p_\Phi(\phi(x')). \tag{2}$$

This general definition unifies seemingly different definitions of saliency encountered in the literature. It relates to salient observations being considered unexpected, rare, or surprising (Garcia et al., 2001; Itti & Koch, 2001; Torralba, 2003; Walker et al., 1998). It also incorporates the notion of saliency considered in decision-theoretic settings (Lingyun et al., 2007; Torralba, 2003), where bottom-up saliency increases with an increase in the information associated with feature vectors (Gao & Vasconcelos, 2007; Rosenholtz, 1999).

Contrary to approaches that determine saliency maps through an explicit data-driven density estimation (Bruce, 2005; Dabkowski & Gal, 2017; Fong & Vedaldi, 2017; Gao & Vasconcelos, 2007; Huang & Gao, 2020; Itti & Baldi, 2005; Jiang et al., 2019; Li & Yu, 2016; Li et al., 2013; Simonyan et al., 2013; Torralba, 2003; Zhou et al., 2016), a closed form expression for saliency can be given once the feature mapping $\phi$ is fixed. When $\phi$ is continuously differentiable, the information content $-\log p_\Phi$ on $\phi(\mathbb{R}^n) \subset \mathbb{R}^N$ over all image features can be obtained from $\log p_X$ through a simple change of variables (Boothby, 1975) from $x$ to $\phi(x)$. The amount of information for the feature vector $\phi(x)$ at every location $x$, and thereby the saliency $S(x)$, is then given by the expression:

$$-\log p_\Phi(\phi(x)) = -\log \frac{p_X(x)}{\sqrt{\det(J_\phi^t(x) J_\phi(x))}} \tag{3}$$

$$= -\log p_X(x) + \frac{1}{2} \log \det(J_\phi^t(x) J_\phi(x)), \tag{4}$$

where $J_\phi : \mathbb{R}^n \to \mathbb{R}^{N \times n}$ denotes the Jacobian matrix of $\phi$, and $\_^t$ indicates matrix transposition.

A crucial implication of this result is that the computation of saliency can be performed based on purely *local* measurements and without the need to refer to previously observed data or any explicit

---

**Algorithm 1** Temporal Saliency in Probabilistic Forecasting

---

**Require:** The probability density function $p_{Y|X}$
**Input:** A fixed $t_{\text{fut}}$ of interest, a sequence of $m$ preceding observed windows $O = [t_{\text{obs}}^1, \ldots, t_{\text{obs}}^m]$, and the behavioral features $X^j$ for every $t_{\text{obs}}^j$
**Output:** The saliency map $S(O)$ over the observed windows

1: **for each** $t_{\text{obs}}^j \in O$ **do**
2:     Compute the feature mapping $\phi(t_{\text{obs}}^j) \leftarrow h(Y|X = X^j)$
3: **end for**
4: Compute the saliency map $S(t_{\text{obs}}) \leftarrow \det(J_\phi^t(t_{\text{obs}})J_\phi(t_{\text{obs}}))$

---

density estimate of these. Following Equation 4, Loog simplifies the definition of the saliency map to

$$S(x) := \det(J_\phi^t(x)J_\phi(x)), \qquad (5)$$

given that a monotonic transformation of the saliency map does not essentially alter the map.

## 3 CLOSED-FORM TEMPORAL SALIENCY FOR PROBABILISTIC FORECASTING

### 3.1 SETUP AND NOTATION

Given the relevance to forecasting tasks involving human behavior, we borrow the notation from the task formulation of Social Cue Forecasting (SCF) (Raman et al., 2021). Let $t_{\text{obs}} := [o1, o2, ..., oT]$ denote a window of consecutively increasing observed timesteps, and $t_{\text{fut}} := [f1, f2, ..., fT]$ denote an unobserved future time window, with $f1 > oT$. Given a set of $n$ interacting participants, let us denote their features over $t_{\text{obs}}$ and $t_{\text{fut}}$ respectively as $X := [b_t^i; t \in t_{\text{obs}}]_{i=1}^n$ and $Y := [b_t^i; t \in t_{\text{fut}}]_{i=1}^n$. The vector $b_t^i$ encapsulates the multimodal cues of interest from participant $i$ at time $t$, such as head and body pose, facial expressions, gestures, etc. The task is to forecast the density $p_{Y|X}$, denoting a distribution over possible future features given an observed sequence $X$ of the *same features*. Here $Y$ and $X$ denote the multivariate random variables associated with the future and observed sequences respectively. While the original formulation of SCF dealt with conversation settings, it is general enough to apply to other tasks such as pedestrian trajectory forecasting, where the features are simply the locations of individuals. In the rest of this work, for simplicity we denote the feature array at an individual timestep $t$ in $t_{\text{obs}}$ and $t_{\text{fut}}$ as $X_t$ and $Y_t$ respectively, even though they correspond to the same feature types. Given $p_{Y|X}$, the focus of this work is to compute the saliency $S(t_{\text{obs}})$ of an observed $t_{\text{obs}}$ towards the future occurring over a fixed choice of $t_{\text{fut}}$.

### 3.2 DEFINING $\phi$ IN TERMS OF THE UNCERTAINTY OVER THE FUTURE WINDOW $t_{\text{fut}}$

The closed-form expression for information-based spatial saliency in Equation 4 makes it explicit that the choice of the feature mapping $\phi$ determines the actual form of the saliency map. To extend the saliency framework to forecasting settings, we formalize the logical implication in Equation 1 by mapping the window $t_{\text{obs}}$ to the differential entropy of the model's predicted future distribution over $t_{\text{fut}}$ conditioned on the observed features $X$. That is, we define $\phi : t_{\text{obs}} \mapsto h(Y|X = X)$, where the conditional differential entropy of $Y$ given $\{X = X\}$ is defined as

$$h(Y|X = X) := -\int p_{Y|X}(Y|X)\log p_{Y|X}(Y|X)dY. \qquad (6)$$

Following Equation 6, our framework for computing the saliency map is summarized in Algorithm 1. Consider that a domain expert selects a specific $t_{\text{fut}}$ corresponding to a high-order semantic behavior they wish to analyse. Examples include a speaking-turn change (Keitel et al., 2015; Levinson & Torreira, 2015) an interaction termination (Bohus & Horvitz, 2014; van Doorn, 2018), or a synchronous behavior event (Bilakhia et al., 2013). Given an underlying forecasting model and looking back in time before $t_{\text{fut}}$, we compute $h(Y|X = X)$ for different *observed multivariate features* $X$ corresponding to different locations of a sliding $t_{\text{obs}}$. The computed differential entropy values are then inserted into Equation 5 to obtain the saliency of *different $t_{\text{obs}}$ locations* towards the future over the chosen $t_{\text{fut}}$.

Differential entropy possesses favorable properties that make it a suitable choice as $\phi$ for computing the saliency map. First, the scale of the forecast density does not affect the resulting saliency map (see Cover & Thomas, Theorem 8.6.4):

$$h(aY) = h(Y) + \log |a|, \text{ for } a \neq 0, \text{ and} \tag{7}$$

$$h(\boldsymbol{A}Y) = h(Y) + \log |\det(\boldsymbol{A})|, \text{ when } \boldsymbol{A} \text{ is a square matrix.} \tag{8}$$

That is, scaling the distribution changes the differential entropy by only a constant factor. So the saliency map resulting from inserting the entropy into Equation 5 remains unaffected since the Jacobian term only depends on the relative change in entropy across different choices of $\boldsymbol{t}_{\text{obs}}$. Similarly, translating the predicted density leaves the saliency map unaffected (see Cover & Thomas, Theorem 8.6.3):

$$h(Y + c) = h(Y). \tag{9}$$

### 3.3 Computing $h(Y|X = \boldsymbol{X})$

To compute the differential entropy of the future distribution, we need the density function $p_{Y|X}$. Typically, this density is modeled as a multivariate Gaussian distribution (Raman et al., 2021; Rangapuram et al., 2018; Salinas et al., 2019a;b). When the decoding of the future is non-autoregressive, the parameters of the distributions for all $t \in \boldsymbol{t}_{\text{fut}}$ are estimated in parallel. In these cases, the differential entropy has a closed-form expression. Assuming a $d$-dimensional predicted Gaussian distribution with mean $\boldsymbol{\mu}$ and covariance matrix $\boldsymbol{K}$, the expression for the entropy of a multivariate Gaussian distribution is given by (see Cover & Thomas, Theorem 8.4.1)

$$h(Y|X = \boldsymbol{X}) = h(\mathcal{N}_d(\boldsymbol{\mu}, \boldsymbol{K})) = \frac{1}{2} \log[(2\pi e)^d \det(\boldsymbol{K})]. \tag{10}$$

When $\boldsymbol{K}$ is diagonal, so that the predicted distribution is factorized over participants and features, we can simply sum the $\log$ of the individual variances to obtain the feature mapping $\phi$. Note that from Equation 10, for a multivariate Gaussian distribution, the differential entropy only depends on the covariance, or the *spread* of the distribution, aligning with the notion of differential entropy as a measure of total uncertainty. (See (Cover & Thomas, Tab. 17.1; Lazo & Rathie, 1978) for closed-form expressions for a large number of commonly employed probability density functions.)

Another common approach for inferring the future density function is the use of probabilistic autoregressive decoders (Ha & Eck, 2017; Raman et al., 2021; Salinas et al., 2019b; Salzmann et al., 2021). Here, one possible decoding approach (Ha & Eck, 2017; Salzmann et al., 2021) involves taking a specific sample $\widehat{Y}_t$ from the density predicted at each $t \in \boldsymbol{t}_{\text{fut}}$, and passing it back as input to the decoder for estimating the density at timestep $t + 1$. Therefore, the density at $t + 1$ depends on the randomness introduced in sampling $\widehat{Y}_t$. Figure 2 illustrates the concept for two timesteps. Here, a single forecast would only output the shaded red distribution for $Y_2$. This precludes the direct computation of $h(Y_1, Y_2)$ that requires the full joint distribution $p_{Y_1, Y_2}$. In such cases, we have two broad options: using a simplifying assumption to retain computational simplicity, or approximating the differential entropy by sampling.

The simpler option is to redefine our feature-mapping as $\phi : \boldsymbol{t}_{\text{obs}} \mapsto \sum_{t \in \boldsymbol{t}_{\text{fut}}} h(Y_t | \widehat{Y}_{<t}, \boldsymbol{X})$, i.e. we sum the differential entropy of the individual densities estimated at each timestep as an approximation of the total uncertainty over the predicted sequence. Note that following the chain rule for differential entropy (see Cover & Thomas, Eq. 8.62), the joint entropy can indeed be written as the sum of individual conditionals. However, in general,

$$h(Y|X = \boldsymbol{X}) = \sum_{t \in \boldsymbol{t}_{\text{fut}}} h(Y_t | Y_{<t}, \boldsymbol{X}) \neq \sum_{t \in \boldsymbol{t}_{\text{fut}}} h(Y_t | \widehat{Y}_{<t}, \boldsymbol{X}). \tag{11}$$

Employing this approximation (of summing the entropies across all timesteps) relies on the observation that for autoregressive decoding, the parameters of the predicted distribution for $Y_t$ is computed as a deterministic function of the decoder hidden state. That is, $Y_t$ is conditionally independent of $Y_{<t}$ given the hidden state of the decoder $\boldsymbol{s}_t$ at timestep $t$. The underlying assumption is that for a well-trained decoder, $\boldsymbol{s}_t$ encodes all relevant information from other timesteps to infer the distribution of $Y_t$. So at inference, despite being a function of the single sample $\widehat{Y}_{t-1}$, the predicted distribution conditioned on $\boldsymbol{s}_t$ provides a reasonable estimate of the the uncertainty in $Y_t$.

The benefit of employing this assumption is that when each $Y_t$ is modeled using a density function that has a closed-form expression for differential entropy (Cover & Thomas, Tab. 17.1; Lazo & Rathie, 1978), every item in the sum can be computed anlaytically, and we obtain a closed-form expression for the saliency map. Apart from a multivariate Gaussian, the other common choice for modeling $Y_t$ is using a Gaussian mixture (Ha & Eck, 2017; Salzmann et al., 2021). While a closed-form expression for the differential entropy of a Gaussian mixture is not known, approximations that approach the true differential entropy can be obtained efficiently (Huber et al., 2008; Michalowicz et al., 2008; Zhang & Luo, 2017) to directly compute the feature mapping $\phi$.

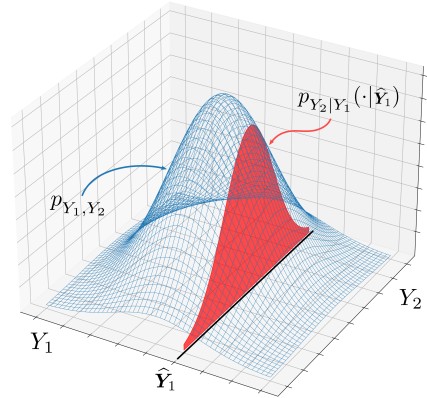

**Figure 2:** Illustrating predicted densities under the greedy autoregressive decoding approach for two timesteps. For simplicity, we depict a joint Gaussian distribution and omit the conditioning on $\boldsymbol{X}$ everywhere.

What if the differential entropy for $Y_t$ does not have an analytical expression or a computationally efficient approximation? In such cases, $h(Y|X = \boldsymbol{X})$ can be estimated using sampling or other non-parametric approaches (Ajgl & Šimandl, 2011; Ariel & Louzoun, 2020; Beirlant et al., 1997; Brewer, 2017). Such sampling-based methods can provide approximations that converges to the true entropy.

## 4 EMPIRICAL VALIDATION OF THE PROPOSED FRAMEWORK

How do we empirically validate that our framework retrieves verifiable salient timesteps towards a future outcome? Crucially, to avoid issues arising from qualitative visual assessment of saliency maps (Adebayo et al., 2018; Barredo Arrieta et al., 2020), a human-verifiable salient relationship between a $\boldsymbol{t}_{\text{obs}}$ and $\boldsymbol{t}_{\text{fut}}$ is needed to serve as ground truth. However, this is non-trivial for real-world data; ground-truthing such saliency in behavior datasets is especially hard because there often exists both supporting and opposing evidence in social psychology for predictive relationships between behaviors (Kalma, 1992; Levinson & Torreira, 2015). So we have a *chicken-egg* problem surrounding evaluation on real-world data: if we had a way of computing such ground-truth predictive saliency, we would not need this research to begin with. Furthermore, even if this is addressed, using real-world data would also entail training a forecasting model on the data. This model is bound to be arbitrarily imperfect, resulting in a correspondingly imperfect saliency map from our framework (garbage in, garbage out). So, when using an imperfect model, do we attribute any flaws in the saliency map to the explainability method or to the imperfect underlying forecasts? This also makes it non-trivial to compare against other gradient and attention-based XAI techniques that require training a specific model to access its parameters. As such, for a fair empirical comparison, any candidate XAI method would need to i. compute saliency by changing observed timesteps one-at-a-time (counterfactual reasoning); and ii. evaluate change in the information in the probablistic forecast to measure the same notion of saliency. No such directly comparable saliency-based explanation frameworks exist for probabilistic forecasts. The closest saliency-based method (Pan et al., 2021) computes saliency maps using random perturbations for non-probabilistic forecasts and requires access to model parameters.

Consequently, our experimental setup is as follows. First, we construct a synthetic dataset of conversation dynamics where a high-order semantic predictive relationship exists between behaviors. Next we assume a perfect underlying forecasting model trained over this dataset: the best a well-trained model can do is to predict the true variance over plausible futures given an observation. Note that the model does not need to have any conception of saliency or high-order semantic concepts. The task is to predict the distribution $p_{Y|X}$ over low-level head pose features denoted by real-valued quaternions, matching the real-world scenario (Barquero et al., 2022; Raman et al., 2021). Our goal is to evaluate if our framework empirically retrieves the timesteps that are salient by construction.

### 4.1 THE DATASET

We choose to synthesize turn-taking dynamics in multi-party conversations, a setting that has received extensive domain interest over the last decade (De Kok & Heylen, 2009; Ishii et al., 2013; 2017; Keitel et al., 2015; Malik et al., 2020; Petukhova & Bunt, 2009; Rochet-Capellan & Fuchs, 2014).

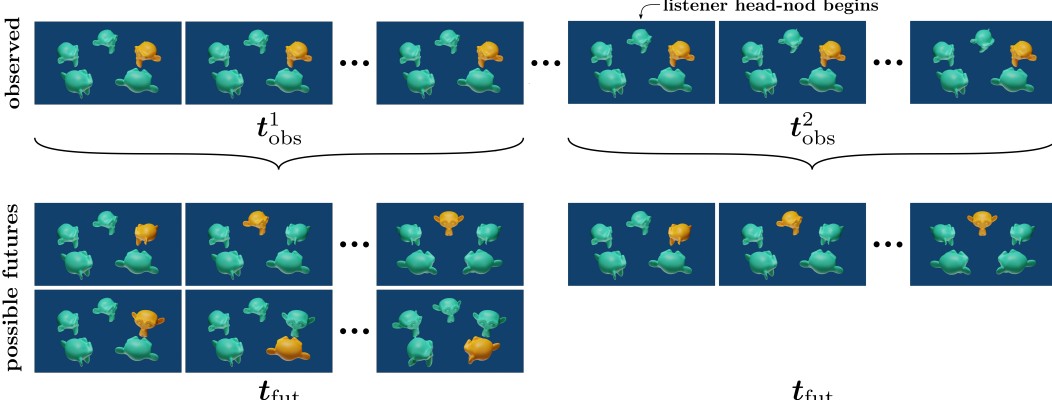

**Figure 3: Illustrating the synthetic conversation dynamics dataset.** Speakers are denoted in orange and listeners in green. For a fixed $t_{\text{fut}}$ we depict two preceding $t_{\text{obs}}$ windows. By construction, when observing a stable speaking turn over $t^1_{\text{obs}}$, two valid futures are possible over $t_{\text{fut}}$. These correspond to a turn handover to the immediate left or right of the current speaker. Over $t^2_{\text{obs}}$, when a listener nods to indicate the desire to take the floor, the future over $t_{\text{fut}}$ becomes certain, corresponding to the listener successfully taking over the speaking turn. Here $t^2_{\text{obs}}$ is consequently more salient than $t^1_{\text{obs}}$ towards forecasting the turn change over $t_{\text{fut}}$.

Specifically, we synthesize a group conversation following established patterns of social behavior. First, the visual focus of attention of listeners is usually the speaker, while the speaker might look at different listeners over the speaking turn (Hung et al., 2008). Second, head gestures and gaze patterns are predictive of the next speaker (De Kok & Heylen, 2009; Ishii et al., 2017; Malik et al., 2020; Petukhova & Bunt, 2009). We implement a simplified version of these dynamics as follows. The speaker rotates towards the geometric center of the formation after acquiring a speaking turn, while the listeners orient towards the speaker. A listener nods their head to indicate a desire to acquire the floor, following which the current speaker rotates to look at the listener and hands over the speaking turn. We represent 3D head poses with quaternions given their common use in representing human motion and pose (Pavllo et al., 2018; Raman et al., 2021). Concretely following the notation in subsection 3.1, $\boldsymbol{b} = [q_w, q_x, q_y, q_z]$. We simulate the turn changes to occur once clockwise and once anticlockwise around the group so that each participant yields the floor once each to the participant on their immediate left and right. We provide code, the dataset consisting of the head-pose and speaking status features, and an animated visualization in the Supplementary material.

## 4.2 COMPUTING TEMPORAL SALIENCY

By construction, for a human observer it is impossible to guess the future with certainty by simply observing the current speaker speaking: there exist two valid examples of the future head behavior in the dataset for such a sequence. Only the commencement of a head nod by a listener makes the future turn handover certain. Figure 3 illustrates this mechanism. Therefore, the timestep corresponding to a head nod is salient towards forecasting the features over a future window involving a turn change in this dataset. Can our framework identify this timestep as salient? A perfect model would predict the true variance over the two valid future quaternion trajectories. This variance vanishes when the future becomes certain as soon as one of the listeners starts nodding. We model the future distribution using a Gaussian function for simplicity (setting std. to $10^{-10}$ for the single future), but a more complex distribution that predicts the appropriate change in variance would also work in practice.

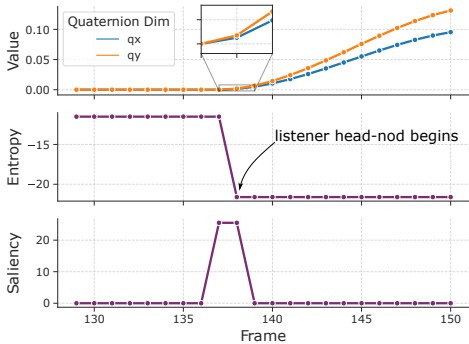

**Figure 4: Computing Saliency.** We plot the quaternion dimensions $qx$ and $qy$ for the listener that nods over $t^2_{\text{obs}}$ in Figure 3 (top). The observation of the head-nod beginning in frame 138 makes the future over frames $183 - 228$ ($t_{\text{fut}}$) certain. This is reflected in the reduction in the mean entropy over future values of all participants (middle). The saliency map obtained using our framework (bottom) correctly identifies the beginning of the head-nod in frame 138 as the salient timestep towards forecasting the future over $t_{\text{fut}}$.

We now implement Algorithm 1 as follows. We identify a window where a turn change occurs in the data (frames 183-228) and denote this 45 frame window as the $t_{\text{fut}}$ of interest. While we manually identify an interesting event for illustration, such a window could also correspond to an interesting prediction by a model. We generate a set of candidate $t_{\text{obs}}$ by sliding a 30 frame window over a horizon of 100 frames prior to $t_{\text{fut}}$, with a stride of 1 frame. For every observed $t_{\text{obs}}$, we fit a Gaussian density to the quaternion and speaking status features of all participants over the futures that can occur during $t_{\text{fut}}$. We then set the entropy of this Gaussian density as the feature $\phi$ for that $t_{\text{obs}}$. We obtain the saliency map using Equation 5, and plot the listener features, entropy, and saliency across timesteps in Figure 4. The ground truth salient timesteps here are frames 138 and 139 where the head nod begins. Our framework retrieves these timesteps as salient with $100\%$ accuracy, empirically validating our framework. Note that once the nod is already in motion, the saliency drops as expected since the future is then already certain given the data. So observing a head nod in motion does not provide any additional information about the future.

## 5 DISCUSSION AND CONCLUSION: SALIENCY & XAI WITH DOMAIN EXPERTS IN THE LOOP

We have proposed a computational framework obtaining counterfactual explanations of model forecasts and established its link to a principled notion of bottom-up task-agnostic saliency. Specifically, we obtain a convenient closed-form expression of the saliency of observed timesteps towards a future outcome for commonly used probability density functions to represent forecasts (Ha & Eck, 2017; Raman et al., 2021; Rudenko et al., 2020; Salzmann et al., 2021). We empirically validate our framework using a synthetic experiment that allows for quantitative validation, avoiding established issues arising from observer biases in visual assessment of saliency maps (Adebayo et al., 2018; Barredo Arrieta et al., 2020). Our goal with this work is to aid domain experts in forming new hypothses from real-world complex data. Given a model that has learned structural predictive relationships from data, our framework identifies timesteps salient towards the model's forecasts through counterfactual reasoning. We thereby envision a human-in-the-loop XAI methodology where the domain expert can form hypotheses surrounding the *features* occurring at these salient timesteps. These hypotheses can then be verified through subsequent controlled experiments.

Loog's (2011) unifying framework subsumes all forms of saliency, although identifying the appropriate $\phi$ for a specific domain is non-trivial. In this work we have established both theoretically and empirically how expressing $\phi$ in terms of the information about the future enables principled counterfactual reasoning in forecasting settings. Nevertheless, we reiterate that the salient timesteps retrieved by our framework ought to be treated as *candidate* causes until subsequently examined along with a domain expert. Our stance on human-in-the-loop XAI also aligns with research on saliency-based and general XAI in other domains (Atrey et al., 2020; Joshi et al., 2021).

In principle, when it is possible to have access to the true $p_{Y|X}$, the salient timesteps identified by our framework reflect the *true* predictive structural relationships captured by the underlying model across the entire data. However, estimating this density analytically entails identifying the multiple futures in the data corresponding to every occurrence of the same observed features. In practice, subtle variations in behaviors and sensor measurement errors make it infeasible to estimate $p_{Y|X}$ analytically, so a model is trained to capture generalized patterns from the given data. In these cases, our framework identifies the sequences that *the model considers salient* for its forecasts *given the data*. Consequently, subsequent causal analysis of the features over the salient timesteps is crucial, especially in the healthcare and human behavior domains to avoid potential prejudices against certain behaviors, or worse, misdiagnoses of affective conditions.

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

# A  BROADER RELATED WORK: EXPLAINABLE METHODS FOR TIME-SERIES TASKS ACROSS DOMAINS

The larger focus of explainability techniques involving time-series data has been on the task of **classifying** time-series. The goal has been to estimate the relevance of each input feature at a given timestep towards each output class. Here, saliency approaches often overlap with techniques developed for image data and can be categorized into:

**Gradient-Based Techniques.** The broad approach involves evaluating the gradient of the output class with respect to the input (Baehrens et al.). Several variants have been proposed (Bargal et al., 2018; Lundberg & Lee, 2017; Shrikumar et al., 2017; Smilkov et al., 2017; Sundararajan et al., 2017).

**Perturbation-Based Techniques.** The idea is to examine how the output changes in response to some perturbation of the input. Perturbations are implemented by either occluding contiguous regions of the input (Mujkanovic et al., 2020; Zeiler & Fergus, 2014); performing an ablation of the features (Suresh et al., 2017); or randomly permuting features (Molnar, 2020). Ismail et al. (2020) provide a benchmark of a subset of these techniques.

**Attention-Based Techniques.** These incorporate an attention mechanism into the model that is trained to attribute importance to previous parts of the input towards an output at each timestep. Such techniques have been specially employed for healthcare data, with early methods applying a reverse-time attention (Choi et al., 2016), and later ones applying the attention to probabilistic state-space representations (Alaa & van der Schaar, 2019).

Some of these broad ideas have been applied to the **regression** setting to make interpretable forecasts of future time-series features. Lim et al. (2020) leveraged self-attention layers for capturing long-term dependencies. Pan et al. (2021) recently proposed computing saliency as a mixup strategy between series images and their perturbed version with a learnable mask for each sample. They view saliency in terms of minimizing the mean squared error between the predictions and ground-truths for a particular instance. Focusing on the univariate point-forecasting problem, Oreshkin et al. (2020) proposed injecting inductive biases by computing the forecast as a combination of a trend and seasonality model. They argue that this decomposition make the outputs more interpretable.

Developing explainable techniques for the probabilistic forecasting setting remains largely unexplored and subject to non-overlapping notions of explainability. Rügamer et al. (2022) transform the forecast using predefined basis functions such as Bernstein polynomials. They relate interpretability to the coefficients of these basis functions (a notion similar to that of Oreshkin et al. (2020)). Panja et al. (2022) embed the classical linear ARIMA model into a non-linear autoregressive neural network for univariate probabilistic forecasting. As before, the explainability here also stems from the 'white-box' nature of the linear ARIMA component. Li et al. (2021) propose an automatic relevance determination network to identify useful exogenous variables (i.e. variables that can affect the forecast without being a part of the time-series data). To the best of our knowledge, saliency-based methods have not yet been considered within this setting.

