# OpenReview forum: "Why Did This Model Forecast This Future? Information-Theoretic Temporal Saliency for Counterfactual Explanations of Probabilistic Forecasts"
_ICLR.cc/2023/Conference — Submitted to ICLR 2023_

### Official Review · Reviewer_bLo2 · 2022-10-17

**Confidence:** 3
**Correctness:** 3
**Technical Novelty And Significance:** 3
**Empirical Novelty And Significance:** 1
**Recommendation:** 3

**Clarity, Quality, Novelty And Reproducibility:**

Clarity: the paper is well written and discussed; however, some practical examples would significantly increase the readability and ease of understanding
Quality: the paper seems to be of good quality; however, my expertise in this theoretical field is limited
Novelty: good, the approach is novel.
Reproducibility: poor; code implementation is not referenced nor discussed, and the synthetic dataset is not publicly available

**Strength And Weaknesses:**

Strengths:
- XAI for time series forecasting is an extremely interesting and underexplored topic
- The authors try to build a formal theory behind saliency maps for time series forecasting


Weaknesses:
- The work is extremely abstract and theoretical and sometimes hard to follow
- The proposed framework is tested qualitatively on only one dataset
- Conclusions are missing

**Summary Of The Paper:**

The authors propose a theoretical framework to explain probabilistic forecasting using a saliency map. They start from Miller's framework to link counterfactual reasoning and saliency-based XAI techniques. Then, they extend the generic notion of saliency to the probabilistic forecasting setting by inferring a closed-form solution for commonly used density functions. The proposed framework is qualitatively evaluated on one synthetic dataset.

**Summary Of The Review:**

The work is extremely abstract and theoretical and sometimes hard to follow. Some practical examples would greatly increase the readability and ease of understanding. The authors should better explain possible applications of the proposed approach and recap their contribution and future work in a Conclusion section.
The proposed framework is tested qualitatively on only one dataset; therefore, it is quite hard to assess its performance and usefulness in real-world applications. The authors should at least provide some quantitative metrics on their synthetic dataset, comparing their framework with other approaches or, if not possible, at least with random or naive approaches.

Other comments:
- page 3: "Our framework addresses all the aforementioned concerns we identified with saliency-based approaches in forecasting: (i) the counterfactuals or alternate 'causes' in our framework pertain to real preceding windows of actual observed features, and are therefore consistent with the underlying data generation process and semantics preserving". To me, this statement is not obvious. Sampling perturbations from past observations does not ensure semantic consistency with future observations. For example, a past observation could only be consistent in certain time series timesteps around specific patterns.
- pag 5, 3.2: "salent" instead of "salient".

In summary, the work is fascinating, but the evaluation part is extremely lacking, and its applicability in real-case scenarios is not adequately discussed.

---

> ### Author Response · Authors · 2022-11-13
> **Thanks for your comments, clarifying and addressing concerns**
>
> Thank you for your time and comments reviewer bLo2. We are glad that you found our work fascinating.
>
> 1\.
> > The work is extremely abstract and theoretical and sometimes hard to follow
>
> We believe that what the reviewer is referring to as the abstract aspects of the work is our conceptual grounding of what constitutes an explanation in established findings in the social sciences. We believe this is needed to address the various current issues in XAI research we have highlighted in Sec. 1, and view this as a strength rather than a weakness.
>
> However, if the paper is hard to follow we would agree with that being a weakness. If the reviewer can be specific about the parts that were hard to follow, we would be happy to try and address those in a revision.
>
> 2\.
> > The proposed framework is tested qualitatively on only one dataset. The authors should at least provide some quantitative metrics on their synthetic dataset $\ldots$
>
> We request the reviewer to kindly refer to our common clarification comment here: https://openreview.net/forum?id=Qi4oCA89CmO&noteId=eczEYHJFw
>
> In the common clarification we discuss how our evaluation is in fact quantitative. We also discuss in detail the broader issues surrounding why our experiment needed to be setup in the way it is, to evaluate our framework independent of any flaws in an underlying forecasting model, and avoid issues arising from qualitative visual comparisons of saliency maps.
>
> 3\.
> >  Reproducibility: poor; code implementation is not referenced nor discussed, and the synthetic dataset is not publicly available
>
> The code, the dataset, a video rendering of the entire dataset, and instructions for running the code were all available in the supplementary material as part of the initial submission itself. We believe this Openreview page is already public, but can move the data and code to Github if needed.
>
> 4\.
> > Sampling perturbations from past observations does not ensure semantic consistency with future observations. For example, a past observation could only be consistent in certain time series timesteps around specific patterns.
>
> What we mean by *semantics preserving* is that the features corresponding to the counterfactuals (different observed timesteps) are real-world features. This is in contrast to the random or masked perturbations other saliency-based techniques apply to the inputs to generate the saliency maps (see Pan et al. 2021). E.g. randomly perturbing head-pose features might result in the head being rotated in ways that would not be possible in the real world, which would break real-world semantics. Alternatively, randomly perturbing timesteps would correspond to glitchy head movements also not possible in the real world. Here such perturbed samples would not lie on the real-world data manifold. We avoid this completely: in our algorithm the 'perturbation' corresponds to adding or removing a contiguous actual timestep to the observed sequence, thereby ensuring that the perturbed sample still corresponds to actual real-world features in the data. For more examples of different types of perturbations, we refer the reviewer to Atrey et al. 2020.
>
> 5\.
> > Examples and usefulness in real-world applications.
>
> The design of the dataset in our empirical experiment is itself meant to serve as an example of how our framework could be used to identify salient timesteps in a real-world task of forecasting low-level future behavior. In Sec. 4.1 we have cited references to illustrate how this task is relevant to the applied domain of social psychology. In the discussion we have further contextualized how our work can be used in aiding domain experts form new hypotheses in a data-driven manner, and how this work relates to human-in-the-loop XAI. We believe the caution we have added in the discussion is also crucial towards how this work will be used.
>
> Please do let us know if the discussion in the common clarification and our response here helps clarify your concerns and how you would like us to address any remaining ones.

---

> ### Author Response · Authors · 2022-11-29
> **Request for discussion**
>
> Dear reviewer bLo2,
>
> We would like to request you for a discussion. We have clarified factual misunderstandings in our responses below and in our common clarification on our experimental design here: https://openreview.net/forum?id=Qi4oCA89CmO&noteId=eczEYHJFw . We have incorporated all your suggestions where possible in our revision as well.
>
> We would appreciate having a discussion since the main remaining complaints the reviewers have had are surrounding adding empirical comparisons that are not straightforward to incorporate in a fair manner either conceptually or operationally.
>
> Best,
>
> Authors

---

### Official Review · Reviewer_YgxY · 2022-10-23

**Confidence:** 5
**Correctness:** 2
**Technical Novelty And Significance:** 2
**Empirical Novelty And Significance:** 2
**Recommendation:** 3

**Clarity, Quality, Novelty And Reproducibility:**

This paper proposes a method to compute temporal saliency in probabilistic forecasting. However, several concepts in this paper are not defined and verified mathematically, which hurt the clarity of this paper. What is more, experiments in this paper are also insufficient.

**Strength And Weaknesses:**


[Strength]
1. This paper focused on an important topic, i.e., explaining the model’s forecast on time-series data.

[Weakness]
1. The introduction of this paper is a little dense, and several concepts are not formulated mathematically, including “associative”, “interventionist”, “counterfactual”, “speculation”, “surprisal”, etc. In the field of explainable AI, researchers are expected to define the mentioned concepts in rigorous mathematical language. Maybe the above concepts are meaningful, but none of them are analyzed mathematically. Therefore, the discussion on these concepts will hardly make sense to readers. Please provide formal mathematical definitions for these concepts, and then discuss their relationship to the proposed method.

2. There lacks evidence to support the claim that “the proposed method is grounded in preattentive human cognition.” First, as mentioned above, please mathematically define components in preattentive human cognition. Second, please provide a theoretical proof to verify that the proposed method indeed aligns with the theory of neuroscience.

3. The actual setting of $\phi$ in Section 3.2 is inconsistent with the introduced physical meaning of $\phi$ in Section 2. In Section 2, $\phi$ is defined as a mapping from image locations to image features. In this case, modeling the distribution of image features as $p_\Phi$ makes sense. It is also meaningful to measure the amount of information using $-\log p_\Phi$. However, in Section 3.2, $\phi$ is defined as a mapping to the entropy of the output. Then, what is the purpose of modeling “the distribution of entropy as $p_\Phi$?” The entropy has its own physical meaning for measuring uncertainty, but modeling the distribution of entropy is meaningless. I suggest authors provide more clarifications on this issue.

4. Authors are encouraged to discuss more about the link between counterfactual reasoning and their method of saliency.
(1) First, I suggest authors mathematically define the “counterfactual explanation” for forecasting settings, instead of only giving a description as “simulating alternative causes to see whether the event still happens” (in Table 1 of Section 1.1). In this way, the connection between the proposed method and counterfactual reasoning can be explained in a more rigorous way.
(2) Second, I do not find experimental results on such counterfactual explanations in Section 4, though authors mention in Section 1.3 that they “examine changes in the future density predicted by M in response to different realizations of real-world sequences $t_{obs}$.” Please conduct some experiments to verify that the proposed method is able to provide counterfactual explanations.

5. Experiments in this paper are insufficient.
(1) No baseline method is compared. Authors are suggested to compare the proposed method with existing widely used attribution methods, including Saliency [cite 1], Occlusion [cite 2], Input x Gradient [cite 3], LIME [cite 4], LRP [cite 5], Shapley value [cite 6], DeepSHAP [cite 7], etc.
(2) Authors also do not verify the effectiveness of the proposed method on benchmark datasets for the time-series forecasting task.

6. About experimental details.
(1) What is the specific form of the future and observed features, $i.e.$, $X$ and $Y$, corresponding to Algorithm 1?
(2) Which one of the three approaches introduced in Section 3.3 is chosen for the experiment to compute the differential entropy?
(3) Moreover, how well is the estimation of the distribution Y|X based on the Gaussian distribution and autoregressive decoders? Please provide a metric to quantify the estimation error, and verify the accuracy of such estimation.
(4) The saliency map in Figure 3 is problematic. Figure 3 shows that the saliency of frames where the nod is not in motion has a zero saliency value. However, the information of the position of the current speaker also affects possible futures. Thus, the saliency of frames where the nod is not in motion may be low, but not zero.
(5) Minor: “Saliency” in the last third line of page 8 is supposed to be “entropy”. According to Figure 3, when the nod begins, the entropy drops, and the saliency increases.

7. About the limitation of the proposed method.
(1) For the time-series forecasting task, the proposed attribution method simply computes the square of the gradient’s L2-norm. Is it equivalent or similar to existing gradient-based explanation methods, such as Saliency [cite 2]?
(2) In Section 2, $\phi$ is restricted to a differentiable function. However, the mapping from image position to feature vector is usually not differentiable, because the image position is a discrete variable. Please provide some clarification for this issue.

[cite 1] Simonyan, Karen, Andrea Vedaldi, and Andrew Zisserman. "Deep inside convolutional networks: Visualising image classification models and saliency maps." arXiv preprint arXiv:1312.6034 (2013).

[cite 2] Zeiler, Matthew D., and Rob Fergus. "Visualizing and understanding convolutional networks." European conference on computer vision. Springer, Cham, 2014.

[cite 3] Avanti Shrikumar, Peyton Greenside, Anna Shcherbina, and Anshul Kundaje. Not just a black box: Learning important features through propagating activation differences. In arXiv: 1605.01713, 2016.

[cite 4] Marco Tulio Ribeiro, Sameer Singh, and Carlos Guestrin. “Why should I trust you?” explaining the predictions of any classifier. In KDD, 2016.

[cite 5] Alexander Binder, Gregoire Montavon, Sebastian Lapuschkin, Klaus-Robert Muller, and Wojciech Samek. Layer-wise relevance propagation for neural networks with local renormalization layers. In International Conference on Artificial Neural Networks (ICANN), 2016.

[cite 6] Lloyd S Shapley. A value for n-person games. In Contributions to the Theory of Games, 2(28): 307–317, 1953.

[cite 7] Lundberg, Scott M., and Su-In Lee. "A unified approach to interpreting model predictions." Proceedings of the 31st international conference on neural information processing systems. 2017.

**Summary Of The Paper:**

This paper proposes a method to compute temporal saliency in probabilistic forecasting. The saliency of each timestamp in time-series data is measured based on its gradient w.r.t. the output entropy. Authors construct a synthesized dataset to verify the effectiveness of the proposed method.

**Summary Of The Review:**

This paper focuses on an important topic. However, several concepts are not formulated in a rigorous way, and some claims are not proved or verified. There are also problems concerning the formulation of the proposed method. Moreover, experiments in this paper are insufficient. Therefore, I recommend rejection.

---

> ### Author Response · Authors · 2022-11-16
> **Thanks for the comments! Clarifications and updates - Part 2**
>
> 5\.
> > Experiments in this paper are insufficient. No baseline method is compared. Authors also do not verify the effectiveness of the proposed method on benchmark datasets for the time-series forecasting task.
>
> Please refer to our common clarification on the matter here: https://openreview.net/forum?id=Qi4oCA89CmO&noteId=eczEYHJFw
>
> Briefly, we explain why fair comparisons against the suggested methods are either conceptually or operationally not possible. Further, there do not exist any benchmark datasets for identifying the salient timesteps towards probabilistic forecasts to our knowledge.
>
> Please, note that the task is  **not** forecasting the future, but identifying timesteps that are salient for a given model's *probabilistic* predictions. As such, this is a new task and ground-truth annoating this is non-trivial for real world data. If the reviewer can expand on how they envision such comparisons without i. resorting to visual assessments of the saliency maps from imperfect underlying models, and ii. obtain ground-truth saliency annotations for real-world data to compare against, we would be happy to incorporate such experiments.
>
> 6\. Clarifications about experiment details:
>
> > What is the specific form of the future and observed features
>
> We represent the 3D head pose at each timestep with a 4D quaternion. We have further expressed this to match the notation in Sec. 3.1 as $\mathbf{b} = [q_w, q_x, q_y, q_z]$ in the revision.
>
> > Which one of the three approaches introduced in Section 3.3 is chosen for the experiment to compute the differential entropy?
>
> In Sec. 4.2 we state that we model the future distribution using a Gaussian function for simplicity (setting std. to 10−10 for the single future), but a more complex distribution that
> predicts the appropriate change in variance would also work in practice. This corresponds to Equation 10 for computing the differential entropy (code is also in the Supplementary material).
>
> > Moreover, how well is the estimation of the distribution Y|X based on the Gaussian distribution and autoregressive decoders
>
> Note that the estimation in the experiment is exact since we have access to the true future distribution from the data, and can thereby evaluate our framework's contribution in isolation, independent of any potential estimation errors from modeling choices of an underlying forecasting model.
>
> > The saliency map in Figure 3 is problematic. Figure 3 shows that the saliency of frames where the nod is not in motion has a zero saliency value.
>
> The saliency is in fact correct: observing a head nod in motion makes the future deterministic, and therefore provides no further information about the future over what the first timestep of the head nod provides. So only the timesteps corresponding to the *beginning* of the head nod. Note that our framework retrieves these simply from the distribution over quaternion trajectories for the entire group.
>
> > Minor: “Saliency” in the last third line of page 8 is supposed to be “entropy”. According to Figure 3, when the nod begins, the entropy drops, and the saliency increases.
>
> Following the previous point, this should still be saliency no? The saliency of the timestps corresponding to the the headnod does drop. We've added "already in motion".
>
> 7\.
> > Is it equivalent or similar to existing gradient-based explanation methods, such as Saliency [cite 2]?
>
> It is not, since (and this is crucial), we are not computing the gradient through the model's parameters at all. The reviewer's reference (Zeiler and Fergus) does not perform counterfactual analysis in the sense of seeing how some output changes in response to changes in inputs, but computes the gradients for a single output wrt to a single input back through the model (which is an associative analysis in nature following Miller (2019)'s framework; corresponding math equations are in Miller 2019 and Pearl and Mackenzie 2018).
>
> The gradients in the Jacobian term in our map does not need access to the internals of the model at all: it computes changes across features of *multiple* predictions (entropy in our case) in response to inputs from *multiple* different windows. So it measures the time steps that are bottom-up salient *to the model*.
>
> > However, the mapping from image position to feature vector is usually not differentiable, because the image position is a discrete variable.
>
> Image gradients are indeed computed by evaluating change in a feature wrt to discrete pixel locations. For instance, Loog (2011) provides examples of the saliency maps when setting $\phi$ to the pixel intensity and color. Also, several classical algorithms like the Canny edge detector use such image gradients wrt discrete image locations.

---

> ### Author Response · Authors · 2022-11-16
> **Thanks for the comments! Clarifications and updates - Part 1**
>
> Thanks for your comments and suggestions, reviewer YgxY! We have incorporated the suggestions we can into the paper, and explain why we believe some of the comparisons requested cannot be operationalized in a meaningful manner for a fair comparison.
>
> 1\.
> > Please provide formal mathematical definitions for these concepts, and then discuss their relationship to the proposed method.
>
> - Please note that the concepts directly related to our framework are all mathematically precise. E.g. we do define surprisal in Equation 2 and connect it to other notions of saliency in Section 2.
> - For a more formal expression of counterfactual analysis we have reworked Section 1.3 in the revision to use causal graphs (Pearl 2009, also "Causal and Counterfactual Inference" by Pearl). We also show how evaluating changes in the predicted entropy in response to different observed sequences relates to saliency through the logical implication, mathematically expressed through our choice of $\phi$ within Loog's general framework.
> - Finally, please note that the mathematical definitions of concepts surrounding the Ladder of Causation are already available in the works we link to (Pearl and Mackenzie 2018, Miller 2019). The current issues in XAI stem not from the lack of mathematical definition of these concepts, but rather a lack of interpretation (or misapplication) of the equations when applied to explanations in models. We identify these implicit interpretations of existing techniques in the introduction, which we believe is a crucial contribution. To keep the work focused, we do rely on the reader to refer to the original works for the equations.
>
> 2\.
> > There lacks evidence to support the claim that “the proposed method is grounded in preattentive human cognition.”
>
> In this work we propose a choice of $\phi$ that enables applying Loog's general mathematical expression of saliency to probabilistic forecasts. In the original paper (Loog 2011 - "Information Theoretic Preattentive Saliency: A Closed Form Solution") Loog discusses at length how his framework already aligns with the notion of preattentive saliency established in human cognition.
>
> 3\.
> > The actual setting of $\phi$ in Section 3.2 is inconsistent with the introduced physical meaning of  in Section 2. Then, what is the purpose of modeling “the distribution of entropy as $p_{\Phi}$?
>
> $p_{\Phi}$ describes the distribution of the features over 'locations': in Loog's original work (Sec 2), these locations are spatial and correspond to pixels. In our work, these locations are temporal, corresponding to where the observed windows $t_{\mathrm{obs}}$ lie in time.
>
> Specifically, we define $\phi : t_{\mathrm{obs}} \mapsto h(Y|X=\mathbf{X})$. That is, to every $t_{\mathrm{obs}}$ we assign as a numerical *feature* the entropy of the corresponding predicted future density. So $p_{\Phi}$ here expresses the distribution of this feature over all locations of $t_{\mathrm{obs}}$ in the data. In Section 1.3 we have argued how the saliency of an observed window in forecasting needs to be *tied to the future outcome*. Mathematically, our $\phi$ achieves this by associating each window with the feature of the model's uncertainty (entropy) over the $t_{\mathrm{fut}}$.
>
> 4\.
> > Authors are encouraged to discuss more about the link between counterfactual reasoning and their method of saliency $\ldots$  Please conduct some experiments to verify that the proposed method is able to provide counterfactual explanations.
>
> We have formally expressed counterfactual analysis using causal graphs in the reworked Sec. 1.3. We have also expressed the logical implication relating such counterfactual reasoning to saliency in the same section. The rest of the paper establishes this link mathematically through defining $\phi : t_{\mathrm{obs}} \mapsto h(Y|X=\mathbf{X})$ within Loog's framework and Algorithm 1. (Also see the arguments building to point 4 in our comment here: https://openreview.net/forum?id=Qi4oCA89CmO&noteId=9Cg4G183W8E)
>
> Our experiment in fact validates that this framework identifies salient timesteps in a counterfactual manner. Note that the Jacobian term in the saliency map in Step 4 of Algorithm 1: $S(t_{\mathrm{obs}}) \leftarrow \det(J^t_{\phi}(t_{\mathrm{obs}})J_{\phi}(t_{\mathrm{obs}}))$ expresses a change in the entropy of the future in response to different choices of $t_{\mathrm{obs}}$ (the counterfactuals). This corresponds counterfactual reasoning using the modified causal graph in Fig. 1b. for different choices of $t_{\mathrm{obs}}$.
>
> The result of the experiment is that our method quantitatively recovers the true salient timesteps with 100% accuracy without any higher-order conception of saliency, thereby validating the framework. (Also please see Q.2 in the linked comment.)

---

> ### Author Response · Authors · 2022-11-29
> **Request for discussion**
>
> Dear reviewer YgxY,
>
> We wanted to request you for a discussion on our response. Where possible, we have incorporated your comments in our revision. Where appropriate, we have clarified misunderstandings in the comments below and in our common clarification on our experimental design here: https://openreview.net/forum?id=Qi4oCA89CmO&noteId=eczEYHJFw .
>
> A short summary:
> Crucially, in our common clarification, we have tried to establish why the comparison against the methods you have suggested are not directly possible either conceptually or operationally. While the suggested methods for comparing against conceptually measure something different than we are, operationally such comparisons would require training a specific underlying forecasting model.  In the absence of ground truth saliency on real-world data, such an experiment would then require visual assessment of the resulting saliency maps known to suffer from observer biases, and make it hard to say whether any flaws arise from our proposed framework or the underlying forecasting model that is independent of our contribution.  Our experimental design was a response to these issues in order to be able to quantitatively validate our theoretical contributions in this paper.
>
> So we would appreciate a conversation since as it stands it is unclear how we can add any such experiments to even a future submission, and we believe we have addressed all your other comments.
>
> Best,
>
> Authors

---

### Official Review · Reviewer_PbXQ · 2022-10-24

**Confidence:** 3
**Correctness:** 2
**Technical Novelty And Significance:** 3
**Empirical Novelty And Significance:** 2
**Recommendation:** 5

**Clarity, Quality, Novelty And Reproducibility:**

As mentioned above, the submission is very clearly written, and is of very high quality over all. The work appears novel, although saliency methods are a bit outside my core area of expertise so it is difficult for me to assess.

**Strength And Weaknesses:**

Strengths:

Overall the paper is extremely well written, and a real pleasure to read. Very nicely done! In particular this submission does a very nice job of arguing why one should use the approach presented, and not just how it works. I suppose we should expect nothing less from a paper about causal explanations.

Weaknesses:

My primary concern with this paper is in the empirical evaluation. While I understand that the problem statement itself is somewhat new, it is still very important to compare the proposed method against (even potentially very weak) baselines. Otherwise, it is impossible to determine whether the proposed method actually delivers on its promise. Additionally, while the figures presented are helpful for their explanatory power, it is very important to somehow quantify performance in terms of some metric. Unfortunately I don't have any concrete suggestions on such a metric.

While it's not necessarily a weakness per se, I am somewhat concerned about the use of the term "counterfactual" here when the authors do not hold exogenous noise constant between factual and counterfactual worlds, as described in (Pearl, 2009) Chapter 7. It would be very helpful for the authors to clarify exactly how the abduction-action-prediction process ala Pearl is reflected in this work. Related, it would helpful to discuss some of the common issues in counterfactual reasoning here, e.g. necessary assumptions for identifiability.


**Summary Of The Paper:**

This submission provides a new information-theoretic approach to inferring salient temporal sequences for counterfactual explanations. Conceptually the authors closely follow the organization framework in (Miller, 2019), partitioning kinds of explanations into analogous categories to the rungs in Pearl's causal hierarchy. The authors go on to present and justify the use of conditional differential entropy as the basis for constructing such a saliency measure. Finally, the authors show the application of the proposed approach on a synthetic conversation dynamics dataset.

**Summary Of The Review:**

Overall the paper is well written, and presents a well motivated and plausible approach to determining temporal saliency. I have some concerns that currently make it challenging to recommend acceptance, but I am also happy to revise my opinion during the discussion.

---

> ### Author Response · Authors · 2022-11-15
> **Thank you for the comments and suggestions, addressing concerns and updates**
>
> Reviewer PbXQ, thank you so much for your time, thoughtful engagement with our work, and for your kind comments! We are happy that you found our paper a pleasure to read.
>
> To address your primary concern surrounding empirical evaluation, we request you to first refer to our broader discussion on the matter in our common clarification here: https://openreview.net/forum?id=Qi4oCA89CmO&noteId=eczEYHJFw
>
> 1\.
> >While I understand that the problem statement itself is somewhat new, it is still very important to compare the proposed method against (even potentially very weak) baselines $\ldots$ I don't have any concrete suggestions on such a metric.
>
> In our common clarification (especially Q2 in the detailed arguments section) we have clarified how our validation is in fact quantitative and equivalent to using accuracy as a metric. The experiment is designed to avoid issues surrounding observer biases or qualitative interpretation of saliency maps. Our framework retrieves the salient timesteps with 100% accuracy in this regard (even though as we explained we were originally hesitant to state it as such in writing). We have now stated it as such in the revision. However, given that it works perfectly empirically and that the rest of our arguments build upon i. well-established findings in the social sciences (Miller 2019), and ii. a general computational framework of bottom-up saliency (Loog 2011), it is unclear what empirical comparison against weak baseline would add or test further. Also, from Q5 in the detailed clarification, it is unclear what a fair simple baseline would even be for an apples-apples comparison.
>
> 2\.
> > It would be very helpful for the authors to clarify exactly how the abduction-action-prediction process ala Pearl is reflected in this work.
>
> Thanks for the suggestion. We have reworked Section 1.3 in the revision to express the explanation setting using causal graphs, and established how it links back to Pearl's abduct-action-prediction process. We hope this addresses the reviewer's comment. Briefly, estimating the distribution over exogenous variables as Pearl describes is not directly required: once the model is trained and the $t_\mathrm{obs}$ and $t_\mathrm{fut}$ have been selected for the counterfactual query the exogenous variables stop influencing the output random variable denoting the information in the model's predicted density.
>
> Also note that the purpose of this work is to aid domain experts in forming new hypotheses about the causal relationships between the *features* occurring over these sequences. This is more related to the problem of causal *discovery*, which pertains to discovering the structure of a different causal graph involving random variables (both endogenous and exogenous) corresponding to the features. Such a graph would be use-case specific however depending on the downstream application.

---

> ### Author Response · Authors · 2022-11-29
> **Request for discussion**
>
> Dear reviewer PbXQ,
>
> We wanted to request you for a discussion on our response. In our revision we have incorporated your suggestions about linking to Pearl's framework on Counterfactuals using an explicit causal graph in Sec. 1.3. We have also addressed your comments below and in our common clarification on our experimental design here: https://openreview.net/forum?id=Qi4oCA89CmO&noteId=eczEYHJFw . We have also incorporated all comments we could in our revision.
>
> We would appreciate having a discussion since, as we have explained in the responses, it is not straightforward to make the suggested empirical comparisons by other reviewers with other methods using real datasets in a fair manner either conceptually or operationally. And so our experiment is designed to quantitatively evaluate our proposed framework, while avoiding issues arising from visual evaluation of saliency maps. As such, it is unclear how we can make the required changes even in a future submission, and would be thankful for a productive conversation with the reviewers.
>
> Best,
>
> Authors

---

### Official Review · Reviewer_7H98 · 2022-11-05

**Confidence:** 4
**Correctness:** 3
**Technical Novelty And Significance:** 2
**Empirical Novelty And Significance:** 1
**Recommendation:** 3

**Clarity, Quality, Novelty And Reproducibility:**

The presentation of the paper is clear and the proposed method is technically sound. However, it lacks rigorous empirical evaluations.

**Strength And Weaknesses:**

Strength:

This paper proposes a reasonable and principle solution to provide salient time steps for time-series forecasting tasks.

It considers a probabilistic forecasting model which is critical for robust predictions.

The proposed method does not need explicit training but obtains a closed-form solution which easy to compute.

Weakness:

The paper lacks a quantitative evaluation of the quality of the salient maps.

The paper did not compare with existing methods, e.g., gradient-based methods and attention-based methods. They are quite straightforward to be extended to the forecasting tasks.

More datasets are expected to show the effectiveness of the method.

**Summary Of The Paper:**

This paper aims to provide salient time steps for probabilistic time-series forecasting tasks. It builds on Miller's framework of explanations and establishes a link between counterfactual reasoning and saliency-based explanation technique; it proposes a unifying definition of information-theoretic saliency for forecasting tasks. Also, the paper empirically shows that the proposed method recovers the true salient timesteps for a synthetic dataset.

**Summary Of The Review:**

This paper proposes a new method to obtain salient timesteps for forecasting tasks. The proposed method is simple and has a closed-form solution. However, the paper lacks rigorous empirical evaluation with existing methods to demonstrate the effectiveness of the proposed method.

[After rebuttal] I do not think the updated draft has addressed the concerns about the empirical evaluation part. Therefore, I keep my original score.

---

> ### Author Response · Authors · 2022-11-13
> **Thank you for your comments, clarifying and addressing concerns**
>
> Thank you for your comments and time, reviewer 7H98! We are glad that you found our solution to be reasonable, principled, and technically sound, and our presentation to be clear.
>
> As context to our specific responses below, we request the reviewer to refer to our common clarification on the design of our empirical evaluation here: https://openreview.net/forum?id=Qi4oCA89CmO&noteId=eczEYHJFw
>
> 1\.
> > The paper lacks a quantitative evaluation of the quality of the salient maps.
>
> Our experiment for empirically validating our framework is in fact quantitative. In particular, please note our comment under Q2. in the detailed clarification section.
>
> 2\.
> > The paper did not compare with existing methods, e.g., gradient-based methods and attention-based methods. They are quite straightforward to be extended to the forecasting tasks.
>
> Following from the arguments in our common clarification (especially Q5 in the details section), we believe it is not straightforward to extend the gradient and attention based methods to handle probabilistic forecasts, or find a reliable way to evaluate the resulting saliency maps since they would require training an underlying model. Besides, this would also be an apples-oranges comparison since they measure a different notion of saliency. Given the larger discussion in the clarification surrounding issues with such comparisons, we request the reviewer to please clarify how they envision such an experiment and reliable evaluation.
>
> 3\.
> > More datasets are expected to show the effectiveness of the method.
>
> We did consider creating more examples in more complex behavior simulators where ground-truth salient timesteps can be established. However, these would all work in the same way as our experiment does and would effectively be the same experiment. We have discussed in our common clarification issues with setting up experiments with real-world data without ground-truth salient timesteps predictive of future outcomes, which we are proposing our framework to help identify in the first place. If the reviewer has any concrete suggestions in this regard that resolves the evaluation issue, we would be thankful to know about them.
>
> Thanks again!

---

> ### Author Response · Authors · 2022-11-29
> **Request for discussion**
>
> Dear reviewer 7H98,
>
> We wanted to request you for a discussion on our response. We have addressed your comments below and in our common clarification here: https://openreview.net/forum?id=Qi4oCA89CmO&noteId=eczEYHJFw . We have also incorporated all comments we could in our revision.
>
> We would appreciate having a discussion since, as we have explained in the responses, it is not straightforward to make the suggested empirical comparisons with other methods using real datasets in a fair manner either conceptually or operationally. And so our experiment is designed to quantitatively evaluate our proposed framework, while avoiding issues arising from visual evaluation of saliency maps.
>
> Best,
>
> Authors

---

### Author Response · Authors · 2022-11-13
**Common clarification: On the design of our empirical validation and issues with the suggested comparisons**

**TL;DR**: Our empirical evaluation is in fact quantitative and designed to avoid well-established issues in explainable AI arising from qualitative or visual comparisons of saliency maps from competing methods. The methods the reviewers have requested comparing against i. cannot be applied to probabilistic forecasts in any straightforward way, ii. do not involve counterfactual reasoning, and iii. require training and accessing the internals of an underlying forecasting model that is bound to be imperfect; this makes it hard to evaluate any flaws of our post-hoc framework in isolation independent of those of the underlying model. Our empirical experiment avoids these issues, and we see no way to include the suggested experiments without propagating the existing issues in XAI.

---

### Preface and Summary

Dear reviewers, meta-reviewer, future readers,

The main concern the reviewers have voiced pertains to our empirical evaluation. This was a central point of discussion amongst the authors in conceiving this work, and in the detailed clarification in the comments below we expand on the broader considerations that led to the experimental setup in the paper (summarized in the introduction to Sec. 4).

Brief recap - as the reviewers have noted, our contributions here are:

* `a reasonable and principled solution to provide salient time steps for (probabilistic) time-series forecasting … that does not need explicit training but obtains a closed-form solution` (Reviewer 7H98)
* we do this by `build[ing] a formal theory behind saliency maps for time series forecasting` (Reviewer bLo2)

We observe that an empirical evaluation can serve the following purposes:

* **validation**: does our closed form solution empirically retrieve timestep(s) salient towards a future outcome when no flaws can be attributed to the underlying forecasting model (garbage forecast in - garbage saliency map out, Sec. 4 introduction)
* **comparison**: how does our proposed method compare against other existing methods for obtaining salient timesteps towards *probabilistic* forecasts for a *given forecasting model*?

As we establish in the comments below, our experimental design is the correct, necessary, and sufficient empirical *validation* of our contributions. We believe the reviewers' comments about the empirical evaluation pertain more to comparisons than validation. Here, the summary of our stance is that neither a fair nor a meaningful comparison against the suggested family of methods is possible (we already do provide the reader with a broader view of the research landscape in Sec. 5 where we summarize Gradient-Based, Perturbation-Based and Attention-Based techniques). Briefly, i. existing gradient or attention based methods cannot in fact be directly deployed for probabilistic forecasts without us inventing a new approach within the corresponding family, and ii. the existing saliency-based methods do not account for probabilistic forecasts and do not capture the same notion of saliency. Besides, any comparative evaluation against the suggested methods would necessitate strawman arguments of improved performance, propagating the existing issues in XAI we have established in Sec. 1.

Our experimental design is the result of considering and addressing the same problem Reviewer PbXQ ran into: `It is very important to somehow quantify performance in terms of some metric. Unfortunately I don't have any concrete suggestions on such a metric.` But also other questions: How do we defend against the issues of biases and interpretation plaguing saliency maps and XAI as other works have shown? What test would evaluate our framework’s contribution without reflecting the imperfections of an underlying forecasting model? Can we actually compare against any baselines on real-world data?

Based on the discussion in the next section, we request the reviewers to consider whether in part the concerns arise from an issue of optics, in that on the surface our paper texture does not match a canonical Experiments section (comparisons against baselines on several real-world datasets, ablations for evaluating components, etc.), especially since these comparisons are not applicable in this situation. At the very least, it would be useful to us, irrespective of the outcome, to have an actionable takeaway from this, because as it stands, we believe that the requested comparisons cannot be made in a reasonable or clear way.

We are grateful for the time the reviewers have already put in, and the positive comments. Thank you all for your time.

Best,

The authors

---

> ### Author Response · Authors · 2022-11-13
> **Detailed Clarification and Broader Discussion - Part 2**
>
> *Q5. Can we meaningfully compare more broadly against other XAI methods?*
>
> We believe not. For a fair apples-apples comparison, any comparable method would need to compute saliency by changing observed timesteps one-at-a-time and evaluate change in the information in the probablistic forecast. That is, it would need to i. measure the same notion of saliency, and ii. compute the saliency map by considering multiple input sequences (counterfactuals) rather than providing a map for timesteps within a single given sequence from a single prediction.
>
> - Techniques where interpretability stems from coefficients of simple basis functions (Oreshkin et al. 2020, Rugamer et al. 2022): The coefficients here cannot be compared one-to-one against a saliency map.
> - Gradient and attention based approaches (@reviewer YgxY, 7H98): There are three issues here:
>     -  The methods either need computing gradients through a specific underlying forecasting model, or accessing the attention weights. In contrast, our method does not require accessing the internal parameters of a model. *If the model is less than perfect, do we attribute any flaws in the saliency map to the explainability method or to the imperfect underlying forecasts?* It may seem like we can simply keep the model fixed and visually evaluate the saliency maps. Here any interpretations would suffer from well-established issues of biases in the observer's assessment (See Sec. 1.1, Adebayo et al. 2018, Miller, 2019).
>     - Further, note that our algorithm works by examining changes over *multiple* predictions corresponding to changing the observed sequence one timestep at a time. The existing attention and gradient based methods yield an attention map over the input timesteps for a single prediction corresponding to a single observed sequence. These measure different things and are not comparable.
>     - Finally, even if we somehow fix these issues, no ground-truth saliency exists in real-world datasets, so it is impossible to compare the resulting saliency maps against a common ground truth (see note on chicken-egg problem in the introduction to Sec. 4).
>
> *Q6. Can we set up useful experiments with real-world datasets?*
>
> We did consider running experiments with real-world datasets, which runs into two problems:
>
> 1. Ground-truthing saliency towards future outcomes in human behavior datasets is especially hard because there often exists both supporting and opposing evidence in social psychology for predictive relationships between behaviors [R1, p.5; R2, p.22]. So we have a *chicken-egg* problem surrounding the evaluation on real-world data: if we had a way of identifying such ground-truth salient predictive relationships, we wouldn’t need this paper.
> 2. Even if we somehow obtain such ground-truth (which is not straightforward), we would then need to train a forecasting model on the real-world data. As discussed above, this would fall into the same issues where validation is based on an imperfect underlying forecasting model. Besides, on real-world data it is impossible to assume a perfect model to isolate any potential issues to our framework. This requires computing the true distribution over futures for any sequence in the data which is not possible for real-world data.
>
> *Q7. What is the broader issue with using human annotation as ground truth for validation here?*
>
> The downstream use of this work is towards aiding domain experts in forming new data-driven hypotheses (Abstract, Discussion in Sec. 6). Here, a forecasting model is assumed to have learned predictive patterns from data that may or may not be already known by experts. By identifying the timesteps that change this model's certainty in a counterfactual manner, we argue that domain experts can hypothesize about potential causal relationships between the features occurring at these timesteps. In Sec. 6 we have cautioned that these hypotheses need verification through subsequent dedicated controlled experiments. Indeed, even knowing when to defer to a domain expert's opinion for explanations is an active line of research (Joshi et al. 2021).
>
> So, in this case, it would be backwards to use a downstream human observer's annotation of complex real-world data (that would need further verification in controlled domain-specific experiments) as ground truth for validating an upstream framework. Our experiment therefore empirically validates our framework in isolation of both, the model being explained and any observer biases.
>
> ---
>
> Consequently, we argue that the empirical experiment we have presented in the paper is sound, more rigorous than relying on qualitative visual assessment, and thereby sufficient for establishing our claims.
>
> ### References not already in the paper
>
> [R1] “Timing in turn-taking and its implications for processing models of language.” - Stephen Levinson and Francisco Torreira.
>
> [R2] “Gazing in triads: A powerful signal in floor apportionment.”- Akko Kalma.

---

> ### Author Response · Authors · 2022-11-13
> **Detailed Clarification and Broader Discussion - Part 1**
>
> We begin by summarizing the paper's arguments to identify which of them need an empirical evaluation:
>
> 1. [Sec. 1.1, 1.2] To move towards causal post-hoc explanations of probabilistic forecasts, counterfactual reasoning is important. *Justification*: Miller's framework grounded in insights from social sciences surrounding what constitutes an explanation.
> 2. [Sec. 1.2] Counterfactual explanations happen to be linked to the broad ideas of saliency-based explanations. *Justification*: In Miller's framework, counterfactual explanations involve simulating alternative causes for an event and evaluating if the event occurs (Table 1).
> 3. [Sec. 1.2] Existing saliency-based explanations: i. suffer from the lack of a consistent definition of saliency (Barredo Arrieta et al. 2020), ii. the perturbations to compute them are often not semantics preserving (Atrey et al. 2020), and iii. the saliency maps are often independent of the data and model (Adebayo et al. 2018).
> 4.
>   - With this, we propose the following implication:
> > [observing the features at timestep $t \in t_\mathrm{obs}$] results in a change in (information about the future over the fixed window $t_\mathrm{fut}$) $\implies$ $t$ is salient.
> >
> (Note that the form of the statement in the paper matched that of the hypotheses in Atrey et al. 2020 to aid the familiar reader. We have since realized that the ordering of the antecedent and consequent ought to be reversed, and have fixed it in the revision coming shortly.) Here $[\ldots]$ denotes the counterfactuals and $(\ldots)$ denote the 'event' in Miller's framework.
>   - To mathematically express the implication, we propose choosing $\phi := h(Y|X=\mathbf{X})$ within Loog's general expression of saliency. This links counterfactual explanations to Loog's general mathematical expression of bottom-up saliency, and addresses some well-established issues with saliency maps as explanations.
>
> *Q1. Which of our arguments need empirical evaluation?*
>
> The claim that choosing $\phi := h(Y|X=\mathbf{X})$ retrieves timesteps that can be considered to be salient towards a given model's forecasts over a given $t_\mathrm{fut}$. The rest of our arguments are built upon established findings from the social sciences (see Miller 2019), or an established unifying framework of saliency (see Loog 2011).
>
> *Q2. Does our empirical experiment validate this claim?*
>
> Yes. The experiment demonstrates that, given the forecasting model's predictions, our framework retrieves the salient timesteps when, crucially,
>
> 1. there are human-verifiable ground-truth salient timesteps that predict a future outcome, and
> 2. the underlying forecasting model is not flawed (defending against garbage in - garbage out, see introduction to Sec. 4).
>
> Note that our experiment uses low-level features typically used in real-world datasets (quaternions for head-pose in this case) and assumes that the forecasting model has no conception of the high-order notion of saliency. The turn-taking behaviors are still grounded in findings in psychology, but the dynamics have been simplified to establish ground-truth saliency towards a future outcome, something that is not available or straightforward to annotate in real-world data (more on this in Q6 and Q7).
>
> @Reviewer PbXQ, 7H98: The experiment's evaluation is in fact quantitative. The metric is accuracy, where the ground truth is a binary vector for the timesteps in $t_\mathrm{obs}$, with a value $1$ for the timesteps salient in predicting the future outcome over the given $t_\mathrm{fut}$, and $0$ everywhere else. Our framework retrieves the salient timesteps with 100% accuracy. In writing, we were hesitant to state this as such, because it seemed a little pretentious to present it as a metric only to state that we obtain 100% accuracy (mathiness for mathiness' sake). Nevertheless, if this makes things clearer, we are happy to describe it as such in the paper and shall update the revision.
>
> *Q3. Does this validation apply to the real world?*
>
> Yes. Note that we needed to use simplified dynamics to establish ground-truth salient timesteps and compute the true future distribution for any observed window in the data. These are needed for validation. The complexity difference between our dataset and real-world data only affects the underlying forecasting model rather than our explainability framework which works only on the output of the model like it does in our experiment. See Q6 and Q7 below for a discussion on what makes correctly evaluating on real-world data a challenge here.
>
> *Q4. What **directly** related explainability methods can we compare against?*
>
> None. As far as we know there is no saliency-based explanation framework for *probabilistic* forecasts. We are the first to pose the question of identifying observed timesteps that are salient towards a probabilistic prediction over a future window from a given forecasting model.

---

### Author Response · Authors · 2022-11-15
**New Revision**

Dear Reviewers,

Thank you all for your comments! We are grateful for the feedback and have improved the manuscript. We have posted a revision of the paper with changes highlighted in blue. Our main changes are as follows:

1. *Reworking Section 1.3*: In response to reviewers PbXQ's comment on linking to Pearl's theory on counterfactuals, and reviewer's YgxY's comment on providing more formal definitions of concepts, we have expressed our approach linking counterfactual reasoning and bottom-up saliency using causal graphs.
2. *Establishing challenges with suggested empirical comparisons*: We have expanded the introduction to Section 4 further explaining why the empirical comparisons suggested by the reviewers cannot be operationalized in a reasonable manner for fair comparisons. This summarizes the discussion in our common clarification comment here:  https://openreview.net/forum?id=Qi4oCA89CmO&noteId=eczEYHJFw
3. *Specifying that the empirical validation is designed to be quantiative*: In response to reviewer statements that our empirical validation is qualitative, we have clarified how our experiment is designed to in fact enable quantitative evaluation.
4. *Summarizing contributions and envisioned use in the Conclusion*: This in response to reviewer bLo2's suggestion. We have also expanded on how we envision the contribution to aid in domain-expert-in-the-loop explainable AI.

---

### Decision · Program_Chairs · 2023-01-20

**Decision:**

Reject

**Justification For Why Not Higher Score:**

All reviewers expressed major concerns and recommended reject, and I am not inclined to override this consensus, based on my reasons stated above.

**Justification For Why Not Lower Score:**

N/A

**Metareview: Summary, Strengths And Weaknesses:**

This paper proposes an information-theoretic saliency-based framework for counterfactual reasoning in probabilistic forecasting, and constructs and validates the approach on a small synthetic dataset. Reviewers agreed it’s an interesting, relevant topic, and found it overall well-written and nicely presented.

However, all reviewers expressed the same major concerns regarding the lack of rigorous empirical evaluations and comparisons to other methods. There was also no demonstration of how this technique can be applied to real-world datasets. Authors emphatically maintained throughout the rebuttal period that they are unable to make any sort of comparisons with other methods, and that it’d be impossible to apply to real-world datasets.

I’m not convinced by their arguments that you can’t compare to other XAI methods. They claim it’s because it’d be an unfair comparison as

>any comparable method would need to compute saliency by changing observed timesteps one-at-a-time and evaluate change in the information in the probablistic forecast.

They then argue that there are many differences between these approaches and theirs. However, it seems like comparison is not impossible, it’s simply imperfect (“It may seem like we can simply keep the model fixed and visually evaluate the saliency maps. Here any interpretations would suffer from well-established issues of biases in the observer's assessment.”) This is not a reason to not do the comparison at all. I’d suggest first performing the comparisons, and then clarifying why these are imperfect comparisons.

I’m also not convinced by their reasoning that they can’t apply this to real-world datasets. They state that

>given the forecasting model's predictions, our framework retrieves the salient timesteps when, crucially,
there are human-verifiable ground-truth salient timesteps that predict a future outcome, and
the underlying forecasting model is not flawed (defending against garbage in - garbage out, see introduction to Sec. 4).

At the same time, they also claim that there are obstacles to applying to real-world datasets because:

>Ground-truthing saliency towards future outcomes in human behavior datasets is especially hard because there often exists both supporting and opposing evidence in social psychology for predictive relationships between behaviors [R1, p.5; R2, p.22]. So we have a chicken-egg problem surrounding the evaluation on real-world data: if we had a way of identifying such ground-truth salient predictive relationships, we wouldn’t need this paper.

However, if this is the case, then doesn’t that mean that their method, which relies on human-verifiable ground-truth saliency, simply can’t be used in practical situations? Doesn’t this then just become a theoretical exercise?

>Even if we somehow obtain such ground-truth (which is not straightforward), we would then need to train a forecasting model on the real-world data. As discussed above, this would fall into the same issues where validation is based on an imperfect underlying forecasting model. Besides, on real-world data it is impossible to assume a perfect model to isolate any potential issues to our framework. This requires computing the true distribution over futures for any sequence in the data which is not possible for real-world data.

Again, just because the metric would be imperfect or would have inaccuracies doesn't mean that one shouldn't try, if only to at the very least demonstrate that the approach is applicable in principle.

I’m sympathetic to the authors’ frustration, but acceptance at ICLR requires at least answering the questions “what do I expect my readers to get out of this?” “How can others apply this to their own work / datasets?” “Why should people apply this approach as opposed to what they’re currently doing to solve this problem?” Once authors reframe the paper to more directly answer these questions, I think they’ll find more receptiveness from reviewers at these types of conferences.

As all reviewers have the same major concerns, I am unable to recommend acceptance.